# Physicochemical, Microbiological and Sensory Characteristics of White Brined Cheese Ripened and Preserved in Large-Capacity Stainless Steel Tanks

**DOI:** 10.3390/foods12122332

**Published:** 2023-06-09

**Authors:** Theofilos Massouras, Evangelia Zoidou, Zinovia Baradaki, Marianna Karela

**Affiliations:** 1Laboratory of Dairy Science, Department of Food Science and Human Nutrition, Agricultural University of Athens, Iera Odos 75, Votanikos, 11855 Athens, Greece; 2Galaktokomiki Kritis Dairy S.A., Selia, 74053 Rethymno, Greece

**Keywords:** white cheese in brine, large-capacity stainless steel tanks, tin containers, physico-chemical characteristics

## Abstract

The objective of the present study was to investigate the effect of ripening and preservation containers on the physico-chemical, microbiological, and textural characteristics, and volatile profile of white cheese. White cheeses were manufactured on an industrial scale using large-capacity stainless steel tanks (SST) of 500 kg, and the respective control samples in tin containers (TC) of 17 kg. No significant differences (*p* > 0.05) in fat in dry matter and total protein content were observed at 60 days of ripening between the TC and SST cheeses. After 60 days, of ripening, the moisture of the cheeses in SST and TC did not show significant statistical differences (*p* > 0.05). No significant differences (*p* > 0.05) were observed between the TC and SST cheeses in the mineral concentration (Ca, Mg, K, and Na) and textural characteristics. Similar results of pH and bacterial counts, as well as absence of yeasts and molds, were observed during ripening and preservation time in both groups of cheeses. Furthermore, proteolysis was not affected statistically significantly (*p* > 0.05). A moderately increased rate of ripening for the cheeses in TC was observed up to 90 days but, at 180 days, proteolysis was similar in both groups of cheeses. Regarding the SFA, MUFA, and PUFA content, no significant differences (*p* > 0.05) were observed between the TC and SST cheeses. A total of 94 volatile compounds were identified in the volatile fraction of both the SST and TC cheeses. Organic acids and alcohols were the most abundant classes of volatile compounds that were identified. The flavor and texture scores in the TC and SST cheeses were similar (*p* > 0.05). Overall, the TC and SST cheeses did not show any significant statistical difference in any of the analyzed parameters.

## 1. Introduction

Different types of cheese have been produced in wide-ranging textures, flavors, and forms, in different regions with unique cultures and environments. Different cheesemaking techniques have been developed over time in response to new technologies and consumer demands. Cheeses can be grouped or classified according to criteria such as the production method, animal milk used, ripening process, country of origin, texture, fat and moisture content, etc. However, key differences in cheese characteristics can generally be attributed to the origin of the milk, moisture content, variation in the container for ripening and preservation, as well as lengths of aging [1].

White brine cheeses (WBC) undergo a ripening process during which they develop microbiological and technological characteristics [1]. WBC are particularly popular in Balkan, Middle Eastern, and Mediterranean regions, and in North Africa and Eastern Europe. They include a large number of varieties, such as Feta (Greece), Telemea or Telemes (Romania, Greece), Akawi (Lebanon, Syria), Halloumi (Cyprus) etc., and are produced by different processing methods, so they have differences in their physico-chemical, textural, and sensory properties [2]. Their manufacture dates back thousands of years (approximately 8000 years ago) [3].

WBC, in general, have a texture that varies from soft to semi-hard. These cheeses have no rind, a slightly acidic taste, due to the action of lactic acid bacteria during ripening, and a salty taste, which arises from storage in brine. Therefore, salt and acid are the critical parameters for the conservation of these types of cheese [4,5]. The main differences among the cheeses are observed in the manufacturing process (for example, milk type, coagulation time and temperature, pressure during draining, shape and size of the curd, and salting of the curd before brining). Sheep or goat milk, or their mixture, is commonly used in their production, however, due to the high purchasing demand, other types of milk, such as cow’s or buffalo’s, can be used [4]. Raw (according to legislation) or pasteurized (72 °C for 15 s or 63 °C for 30 min) milk cheese is used in WBC manufacture [6]. Thermophilic or mesophilic lactic acid bacteria, or combination of them, are commonly used as starter cultures. Non-starter lactic acid bacteria (NSLAB) are predominant when using unpasteurized milk in production.

WBC are white in color, except for those made from cow’s milk that gives a yellowish color due to the presence of carotenoids. These cheeses have no gas holes but they sometimes develop small mechanical openings [3]. The texture of WBC is smooth, soft, and crumbly but still sliceable, and some of them may become brittle when old [6]. Their flavor is slightly sour to very salty and, for some varieties, mildly acidic and piquant. The shape varies but is usually produced in rectangular or cubic blocks that weigh 250–1000 g. The cheeses are packed in containers of various capacities. The most common are rectangular tin cans, lacquered metal or plastic containers with up to 15–16 kg capacity or, traditionally, wooden barrels of 40–50 kg capacity [3,6]. Most WBC are consumed fresh after ripening for 2 months or more in brine (8–10% NaCl concentration) [3,6]. Post-ripening, WBC are repacked in plastic bags under vacuum (without brine) or in plastic containers with brine [6].

Tin cans are usually used for the ripening and preservation of WBC, before their sub-packaging in smaller containers. However, in recent years, stainless steel tanks of large capacity have been used more and more, mainly in large cheese factories. More specifically, the use of tin containers (TC) was one of the most suitable for the ripening, packaging, and handling of brine cheeses due to their low cost, light weight per unit area, easy handling, and O_2_ product protection. Over the years, TC have been partially replaced by other materials, such as plastic and stainless steel. The latter is resistant to corrosion and low temperatures, accepts electro-polishing, and offers the product protection from O_2_, light, odors, and microorganisms. An advantage of the stainless steel tanks (SST) is their reusability and ease of cleaning, even with a CPI system. Thus, the use of SST containers is constantly increasing and mainly by cheese factories, where a large part of their production is intended for repackaging.

Although there are many research studies that refer to the characteristics of WBC matured and preserved in tin or wooden containers, no research to date provide data on the characteristics of cheeses that are ripened and preserved in large capacity stainless steel tanks (SST) (approximately 500 kg).

Considering the growing interest in the use of large capacity containers for ripening and preservation, this study aims to investigate the use of stainless steel tanks (SST), and their effect on the quality of WBC, comparing this to those ripened in tin containers (TC). The physico-chemical, microbiological, textural, and sensory characteristics of WBC, made from sheep’s milk, were studied during their preservation over 180 days in SST and TC.

## 2. Materials and Methods

### 2.1. Cheese Manufacture

The production of the WBC was carried out on an industrial scale, using 5000 L of sheep’s milk, according to the procedure described in Figure 1. Sheep’s milk, with an average composition of 5.78 ± 1.06% fat, 4.87 ± 0.62% proteins, and 4.02 ± 0.55% lactose, was pasteurized in a plate heat exchanger at 72 °C for 10 sec and cooled down to 35 °C. Commercially available freeze-dried direct vat set (DVS) cultures, containing *Lactococcus lactis* (25%), *Lactococcus lactis* subsp. *cremoris* (15%), and *Streptococcus thermophilus/Lactobacillus delbrueckii* subsp. *bulgaricus* (60%) (Chr. Hansen, Copenhagen, Denmark), were used as starter cultures. CaCl_2_ solution (10% *w*/*v*), at a rate of 100 mL/100 kg milk, and powdered calf rennet (HA-LA, Hansen’s Laboratorium, Copenhagen, Denmark) were then added to achieve coagulation in approximately 40 min. The cheese curd was cut into small cubes (2.0 cm^3^), allowed to rest for 10 min and was transferred into rectangular plastic multi-molds (capacity of 2–2.5 kg drained curd). After draining (a day after) the curd was placed into two different ripening–preservation containers; TC (17 kg capacity) and SST (500 kg capacity), respectively, for 180 days. Preservation in TC was used as a control. Three individual replicates were carried out. It is important to mention that the brine that was used had a salt concentration of 7%, with a pH of 7, and a CaCl_2_ concentration of 0.02%.

### 2.2. Cheese and Brine Samples

WBC samples were taken from TC and SST containers and analyzed at 1, 10, 60, 90, and 180 days for physico-chemical composition, microbiological, textural, and sensory characteristics. To establish the interaction between the cheeses and the respective brines, the determination of major minerals composition (calcium, magnesium, potassium, and sodium), as well as the total protein and nitrogen fraction content, of the brines was carried out. All samples were analyzed in triplicate.

### 2.3. Analytical Methods

#### 2.3.1. Physico-Chemical Analysis

The pH was measured using an electronic pH meter (Orion 720 A, Orion Pacific Pty Ltd., Frankston, Vic, Australia). Total solids content was determined by oven drying in a laboratory oven at 105 °C for 24 h, according to the method of I.D.F. [7]. Fat content was measured according to the volumetric method of Gerber [8]. Chloride content determination was performed by the potentiometric titration method according to the I.S.O. [9], and the ash content was determined by dry ashing the samples in a muffle furnace at 550 °C for 24 h, according to the method of I.D.F. [10]. The major minerals (Ca, Mg, K, and Na) concentrations of the samples (both cheeses and respective brines) was determined by Atomic Absorption Spectrometry on a Shimadzu AA-6800 atomic absorption spectrophotometer (Shimadzu AA-6800, Kyoto, Japan) equipped with the autosampler Shimadzu ASC-6100 and the software Wizard v.2.30., according to the method of I.D.F [11].

The extent of proteolysis of the cheeses during ripening was monitored by measuring the levels of total nitrogen (TN), soluble nitrogen (SN) fractions (i.e., water-soluble nitrogen) (WSN), and nitrogen soluble in 12% trichloroacetic acid (TCA-N) or in 5% phosphotungstic acid (PTA-N)), using the Kjeldahl nitrogen determination method [12]. Total protein was calculated by % total N × factor of 6.38. The RP-HPLC peptide profiles of SN, TCA–N, and PTA-N were conducted based on the methodology of Nega and Moatsou [13].

#### 2.3.2. Fatty Acids Composition

For fatty acids (FAs) composition analysis, lipid extraction was performed with solvents after suitable preparation of the samples (WBC) according to I.D.F. [14]. The fat residue extracted was stored in amber vials, exposed to a stream of N_2_ and frozen at −20 °C until analysis. Fatty acids were methylated according to Massouras et al. [15] with some modifications. Briefly, cheese lipid extract of 100 mg was methylated in a screw-cap Pyrex culture tube with the addition of 2 mL of 0.5 M sodium methylate at 50 °C for 30 min, followed by 2 mL of 140 g L^−1^ boron trifluoride in methanol (BF3) at 50 °C for 30 min. Fatty acid methyl esters (FAMEs) were recovered in hexane (2 mL). Each sample (1 μL) was injected by Shimadzu GC-2014 GC AOC-20i autosampler into a Shimadzu gas chromatograph (model GC-17A, Columbia, MD, USA), equipped with a flame ionization detector (FID), and analyzed in duplicate. Separation of fatty acid methyl esters was achieved on a SP-2560 fused silica capillary column (75 m × 0.18 mm I.D., 0.14 μm; Supelco Inc., Bellefonte, PA, USA). Helium (purity N5) was used as a carrier gas with a flow rate of 1 mL·min^−1^ at a split ratio of 1:50 with constant flow control. The injection and detector temperatures used were 250 °C and 270 °C, respectively. The oven temperature program was as follows: the initial temperature was held at 75 °C for 5 min after injection, then programmed to increase at 5 °C/min to 150 °C, to hold for 5 min, and then to increase to 220 °C at 7 °C/min and hold for 20 min. Fatty acid peaks were recorded and integrated using a Shimadzu GC solution software (Shimadzu Corporation, Kyoto, Japan). Individual fatty acids were identified by their retention times and their comparison with known fatty acid methyl ester standards (Supelco 37 Component FAME Mix, purchased from Sigma–Aldrich, Taufkirchen, Germany). Amounts of fatty acids were expressed as a weight percentage of total methyl esters of fatty acid (g·100 g^−1^ of total FAMES).

#### 2.3.3. Analysis of Volatile Compounds by Solid-Phase Microextraction (SPME) and Gas Chromatography–Mass Spectrometry (GC–MS)

Volatile compounds of WBC at the 60th, 90th, and 180th day of ripening were determined using solid-phase microextraction (SPME) combined with GC/MS. Cheese samples (4 g) were homogenized with 2 mL of saturated Na_2_SO_4_ aqueous solution and 100 µL of an internal standard (IS) aqueous solution containing 0.77 g L^−1^ cyclohexanone (Sigma–Aldrich Quνmica, Alcobendas, Spain). Aliquots (3 g) of the homogenates were placed into 22 mL vials sealed with PTFE/silicone septa (Supelco, Bellefonte, PA, USA) through which the SPME syringe needle (bearing a 50/30 μm DVB/CAR/PDMS fiber Supelco, Bellefonte, PA, USA) was introduced. The samples were stirred continuously on a stir plate revolving at 750 rpm. Fiber was exposed to the headspace above the sample for 30 min at 65 °C. The absorbed volatiles were immediately desorbed at 250 °C for 3 min, in splitless mode, into the injection port of a GC–MS system (Shimadzu GC-17 A, MS QP5050). Volatile compounds were separated by a capillary column HP-INNOWax 60 m, 0.25 mm i.d., 0.25 μm film thickness (J&W Scientific, Agilent Technologies Palo Alto, CA, USA). The temperatures for the ion source, quadrupole, and interface were set at 230, 150, and 280 °C, respectively. The oven temperature was held at 45 °C for 5 min, increased to 150 °C at a rate of 5 °C min^−1^, then raised at 7 °C min^−1^ to 220 °C, and held at 250 °C for 20 min. Helium was used as carrier gas at a flow rate of 1.0 mL min^−1^. Electron impact ionization of MS was used at a voltage of 70 eV with a scan range from 40 to 500 *m*/*z*. The volatile compounds were identified by comparing their spectra with those from the NIST (National Institute of Standards and Technology, Gaithersburg, MD, USA) MS library. The volatile compounds were quantified by dividing the peak areas of the compounds of interest by the peak area of the IS, multiplying this ratio by the initial concentration of the IS (expressed as ppm). The peak areas were measured from the full scan chromatograph using total ion current (TIC).

#### 2.3.4. Texture Profile Analysis

Textural profile analysis of the cheeses was assessed with a Shimadzu testing instrument, model AGS-500 NG (Shimadzu Corporation, Kyoto, Japan) equipped with a 5 kg load cell. A plunger with a diameter of 6 mm was attached to the moving crosshead. The speed of the crosshead was set at 2.5 cm mid in both upward and downward directions. The cheese sample was placed on a flat holding plate at 20 °C and the plunger was inserted 20 mm below the cheese surface. Two consecutive bites were taken. The analysis was conducted as described by Kaminarides and Stachtiaris [16]. The following six textural parameters were calculated: Hardness (N), defined as the peak force (H) during the first compression cycle (first bite), is the force necessary to attain a given deformation. Cohesiveness (N mm), defined as the ratio of the positive area under the curve during the second compression to that during the first compression. Adhesiveness (N mm), defined as the negative force area for the first bite, is the work necessary to overcome the attractive forces between the surfaces of the cheese and the plunger with which the cheese comes into contact. Elasticity (mm), defined as the ratio of the base line of the positive curve during the second compression to that during the first compression, is the height that the cheese recovers during the time that elapses between the end of the first and the start of the second bite. Gumminess (N), which is the product of hardness X cohesiveness, is the energy required to disintegrate the cheese to a state ready for swallowing. Chewiness (N), which is the product of gumminess X elasticity is the energy required to masticate a cheese to a state ready for swallowing [16].

#### 2.3.5. Microbiological Analysis

Samples of curd (1 day after the draining), cheeses, and brines at different ripening and storage times (10, 60, 90, and 180 days) were examined for total viable count (TVC), yeasts, and molds, following the I.D.F. [17] and I.D.F. [18] methods, respectively. All the counts were expressed as colony-forming units per gram of cheese (CFU g^−1^).

### 2.4. Sensory Evaluation

Sensory evaluation of the WBC samples was carried out at 60, 90, and 180 days of storage by a trained taste panel of the Dairy Laboratory of the Agricultural University of Athens. Panel members evaluated the cheeses for appearance, flavor, and body-texture using a 10-point scale. More importance was given to flavor and to body/texture than to appearance of the cheese, as advised by I.D.F. (1997) [19], by multiplying their scores by five and four, respectively. Total score was obtained by the addition of scores of the three attributes. Excellent cheese received a total score of 100.

### 2.5. Statistical Analysis

Physico-chemical, microbiological, and sensory parameters of two groups of cheese were subjected to analysis of variance (ANOVA) using Statgraphics Centurion XVII software (Statpoint Technologies, Inc., Warrenton, VA, USA). The difference of the means of the results of the analyses for each component was checked separately by the method of the least significant difference at a significance level of 95% (LSD, *p* < 0.05). The model used was:Yi = μ + Treatment (Ti) + ei
where:

μ = the meanTi = the fixed effect of treatment with i = 1: SST, 2:TCei = the random error, assumed to be normally and independently distributed withzero expectation and common variance.

## 3. Results and Discussion

### 3.1. Physico-Chemical Composition

The results of the physico-chemical analysis of the experimental cheeses and brines during the ripening–preservation period are shown in Table 1 and Table 2. The type of containers (TC and SST) did not have a statistically significant (*p* > 0.05) effect on the content of total solids, moisture, fat, protein, ash, salt, and pH between both the WBC and the brines of the two groups (TC and SST).

After 10 days of ripening, the pH of the cheeses kept in TC and SST decreased, at 4.43 and 4.45, respectively, which is desirable. At 60 days onwards, both white brine cheeses (kept in TC and SST) had the desired pH (up to 4.5). The same pH (<4.6) was observed in the respective brines too. This pH value is particularly important in order to prevent microbial spoilage.

At 60 days onwards, when the cheese is considered ready for consumption, the average moisture content of the cheeses being ripened in TC and SST ranged from 53.04 to 56.34% and 54.13 to 54.63%, respectively. Both cheeses (TC and SST ripened), are classified as quality A, as set out in Greek legislation [20].

During the ripening–preservation process, the fat content (expressed on dry matter) of both cheeses was similar. At 60 days, the fat content of TC- and SST-ripened cheeses ranged from 51.64 to 53.50% and 52.18 to 53.74%, and that remained constant up to 90 days. Based on their fat content, the cheeses from both groups are classified as quality A based on Greek legislation [20].

The protein content (expressed on dry matter) of both TC- and SST-ripened cheeses was similar (*p* > 0.05); during ripening–preservation, and after the 60th day, this ranged from 34.26 to 35.84% and 34.80 to 36.24%, respectively. Both groups presented their maximum value at day 90. In addition, the protein content of the respective brine, was seen to increase continuously up to day 90, when a maximum value for both brines was recorded. This probably indicates the transfer of proteins from the cheese to the brine.

The salt concentration in both cheese groups increased over time and, on the 60th day onwards, ranged from 2.77 to 3.19% and 2.90 to 3.24% in TC and SST, respectively. No significant statistical differences (*p* > 0.05) were observed in the salt concentration of SST-ripened cheeses compared to the TC cheeses. This can be confirmed by the salt content of the brine which, in both cases, decreased up to the 90th day.

The ash content of the cheeses (in TC and SST) was similar (*p* > 0.05) and increased over time and varied from 3.29 to 3.74% and 3.44 to 3.81% for TC and SST, respectively. The physico-chemical characteristics and the pH of TC- and SST-ripened cheeses, throughout their ripening–preservation process, agree with those described in recent research works [21,22,23,24].

### 3.2. Major Mineral Composition

Table 3 and Table 4 show the composition of the major minerals (Ca, Mg, K, and Na) in TC-/SST-ripened cheeses, and the respective brines, during their ripening–preservation. The different ripening–preservation containers did not statistically significantly (*p* > 0.05) affect the concentration of mineral in both cheese groups and the respective cheese brines throughout the ripening–preservation process.

From the 60th day onwards, the concentration of Ca and K was similar, for TC (223.96–313.67 mg 100 g^−1^ and 48.65–86.95 mg 100 g^−1^) and SST (235.05–280.59 mg 100 g^−1^ and 59.42–75.73 mg 100 g^−1^). The concentration of Mg did not have significant differences between the TC and SST cheeses. This was observed throughout the ripening and preservation of the cheeses, as shown in the Table 3. The concentration of Na was similar for both groups of cheeses (1053.40–1215.50 mg 100 g^−1^ for TC cheeses and 1167.88–1266.06 mg 100 g^−1^ for SST cheeses) with the highest being noted at day 180, as was also observed with the salt concentration (Table 1).

The composition of the inorganic elements in both TS- and SST-ripened cheeses, throughout the ripening–preservation process, agrees with previous findings. The amount of Ca and Mg found in TC and SST cheeses were slightly lower than those reported by Abou Jaoude et al. and Barać et al. [25,26], while the concentration of K and Na agreed with those reported by Barać et al. [25].

Regarding the brines, the concentrations of Ca and Mg were similar in both groups (with the exception of the 10th day, when Ca was statistically significantly higher in SST, *p*-value = 0.042). In the TC brine, Ca and Mg ranged between 372.33 and 450.60 mg 100 g^−1^ and 27.85 and 29.91 mg 100 g^−1,^ respectively, while, in SST brine, Ca and Mg ranged between 354.97 and 413.09 mg 100 g^−1^ and 24.60 and 28.42 mg 100 g^−1^. The concentration of K ranged from 104.80 to 120.09 and 92.73 mg 100 g^−1^ to 130.12 for TC and SST brines, respectively, with the highest values being noted on the 180th day. Finally, the concentration of Na was similar in the two groups of brines throughout the ripening–preservation process, with values ranging from 2046.38 to 2255.00 mg 100 g^−1^ and 2234.93 to 2371.95 mg 100 g^−1^ for TC and SST brines, respectively.

### 3.3. Proteolysis

Proteolysis is the most important biochemical event during the ripening of most rennet-coagulated cheese varieties, with a major impact on flavor and texture. During the ripening of the cheeses, caseins are hydrolyzed, resulting in water-soluble nitrogenous fractions (WSN). These fractions are indicators of the proteolysis that the cheeses undergo, which describes the speed and manner of ripening of the cheeses. Proteolysis, in terms of both SN and low-molecular-weight nitrogen fractions (i.e., TCA–SN and PTA–SN) expressed on cheese weight, is lower in brined cheeses compared to semi-hard and hard cheeses [10]. The main reason is the higher moisture content (53–56%) compared to the other cheeses groups and the very high salt-in-moisture content. Furthermore, the migration of whey proteins and soluble proteolysis products into the brine limits the concentration of these products in the cheese mass [10]. Table 5 shows in detail the course of the WSN/TN, TCA-N/TN, and TCA-N/WSN indices of the cheeses ripened and preserved in TC and SST containers. The water-soluble nitrogen ratio (WSN/TN) increased from 8.19% on day 1 to 10.29% and 10.41% for (TC) and (SST), respectively, on day 10, and remained at these levels until, on the 180th day of preservation of the cheeses, they showed small, but not statistically significant, differences (*p* > 0.05). No significant difference (*p* > 0.05) was found for ripening index between the two groups of cheeses during ripening–preservation. The values of the WSN/TN index found for both the TS and SST groups of cheese during the ripening–preservation process agreed with those reported by Zoidou, et al. and by Abd El Salam and Alichanidis [22,27]. The TCA-N/TN index showed a similar trend and no significant differences (*p* > 0.05) between the two groups of cheeses. The TCA-N/TN values in the TS and SST cheeses are in accordance with those of other research studies [22,27]. Finally, the TCA-N/WSN index showed no statistically significant differences (*p* > 0.05) between the two groups during ripening–preservation. Therefore, considering the above, we can draw the conclusion that there was a slightly increased ripening speed in the TC cheeses, up to 90 days; however, at 180 days, the level of proteolysis of both groups of cheeses was similar.

Figure 2 and Table 6, Table 7 and Table 8 show the evolution of the characteristic regions of the RP-HPLC profiles. Chromatograms show some free amino acids and non-nitrogenous soluble components (eluted in the 0–10 min interval), followed by small peptides and the majority of free amino acids (proteolysis products) (in the 10–40 min interval), the water-soluble components (in the 40–70 min interval), and, finally, the hydrophobic components, large peptides, and whey proteins (in the 70–100 min interval). Comparing the chromatographic analysis of the nitrogen fractions of the cheeses and brines in the two ripening-preservation media (TC and SST), no differences were found between them. Specifically, the percentages of the chromatographic surfaces in the time intervals 0–10, 10–40, 40–70, and 70–100 min, of the chromatograms and the ratios (55–100 min)/(10–55 min), and (70–100 min)/(0–70 min) of the nitrogen fractions of cheeses (WSN/TN and TCA-N/TN Table 7 and Table 8, respectively) and brines (Table 6), were similar during ripening–preservation in TC and SST, and did not present any statistically significant difference (*p* > 0.05).

At the 70–100 min range, hydrophobic peptides and whey proteins were eluted. Much of the nitrogenous components of the brine consist of such components (20–30%) throughout the ripening of both cheese types. This shows how whey proteins diffuse into the brine. However, more than 50% of the brine peptides appear to be composed of the soluble components of the 40–70 range, following the course of their increase in the cheese. Therefore, the level of proteolysis and the rate of ripening were similar in TC and SST cheeses.

### 3.4. Fatty Acids Profile

Individual fatty acids and the proportion of fatty acid groups (saturated, SFA; mono-unsaturated, MUFA; and polyunsaturated, PUFA) found in both the TC and SST WBC, at the 60th, 90th, and 180th day of ripening–preservation, are shown in Table 9. The most abundant FAs throughout the ripening–preservation period were palmitic acid (C16:0), oleic (C18:1n-9), and myristic (C14:0) acids. The values for palmitic acid and oleic acid for TC cheeses ranged from 30.74 to 31.14% and 19.14 to 20.72%, respectively, while, for SST, they ranged from 30.24 to 31.08% and 19.14 to 20.72%, respectively. Myristic acid, the third most abundant FA, showed a statistically significant difference between the TC and SST cheeses on day 60 (*p*-value = 0.027) and day 90 (*p*-value = 0.017). For the other identified FAs, similar values with no statistically significant differences (*p* > 0.05) were found between the two groups of cheeses, with the exception of the caprylic acid which presented a statistically significant difference (*p*-value = 0.038) on day 90, with SST having a higher value than TC. Overall, the values of the cheeses, regarding their SFA, MUFA, and PUFA content, were similar between the TC and SST cheeses, and no statistical difference was observed between them (*p* > 0.05). Therefore, we can claim that the different ripening–preservation containers did not affect their fatty acid content. The fatty acid composition reported here was similar to that reported by other studies concerning white brine cheeses [21,23,28].

### 3.5. Volatile Compounds

The results of the volatile compounds analysis of WBCs are shown in Table 10. A total of 94 volatile compounds were identified and grouped into the following chemical classes: organic acids (15), alcohols (16), aldehydes (10), esters (13), ketones (11), lactones (5), terpens (6), alcanes (8), and amines (3). Organic acids, alcohols, and esters, both in number and amount, are the dominant chemical groups. Regarding the individual identified volatile compounds, no significant differences were observed between the TC and SST cheeses. Organic acids were the most abundant chemical class in both cheeses (TC and SST) with acetic acid, decanoic acid, hexanoic acid, and octanoic acid representing about 78% of the total amount of organic acids. Each of them gives a characteristic flavor note [21]. In general, fatty acids, having between 4 and 20 carbon atoms, are formed through lipolysis by microbial lipases. The shorter fatty acids may come (originate) from the degradation of lactose and amino acids, as well as from the oxidation of ketones, esters, and aldehydes [21,29]. As expected, the samples (TC and SST) had the highest content of total organic acids on the 180th day (3828.29 mg/kg for TC and 3846.02 mg/kg for SST). From the chemical class of alcohols, the major alcohol was ethanol (both in TC and SST) throughout their ripening–preservation period, with the maximum value in both cases being noted on the 180th day (530.41 and 549.44 for TC and SST, respectively), which was expected, considering that its formation is due to lactose fermentation, proteolysis, and reduction in acetaldehyde [21,29]. Esters can be produced enzymatically or chemically through the reaction of short to medium chain fatty acids with primary and secondary alcohols that both derive from lactose fermentation and amino acid catabolism [21,30].

Regarding the chemical class of esters, it was observed that their total amount increased with time, with the maximum value being noted on the 180th day in both TC- (220 mg/kg) and SST- (275.48 mg/kg) ripened cheeses. The esters detected in large quantities in all cheese samples were acetic acid 2-phenylethylester, decanoic acid ethylester, and 2-ethenyloxyethanol. Among the less abundant groups, alkanes and terpenes, lactones, and ketones were more represented in both cheeses at 180 days. The volatile profile of white brine cheese in this study is close to those reported by other studies concerning this type of cheese [21,31].

### 3.6. Texture Profile Analysis

Table 11 shows the textural characteristics of the TC- and SST-ripened cheeses during the ripening–preservation period. The ripening–preservation containers did not significantly (*p* > 0.05) affect the textural characteristics of the cheeses, as their values had been similar over time.

The hardness of both groups of cheeses increased with time and specifically in the period of 60–180 days. This increase was enhanced by the progressive increase in the salt and ash content of the respective cheeses during the 60th–180th days [16], and the decrease in the fat content during the 90th–180th days [32]. Both cheese groups showed the lowest (6.04 ± 1.90 N for TC and 7.14 ± 2.47 N for SST) and the highest (9.66 ± 1.10 N for TC and 8.45 ± 2.29 N for SST) values on the 60th and 180th days, respectively. Due to the increase in hardness, the gumminess and chewiness of both cheeses also increased during their ripening–preservation. The cheeses (TC and SST) showed the highest value in gumminess and chewiness on day 180 (3.44 ± 0.37 N and 3.67 ± 0.26 J for TC, and 2.99 ± 0.86 N and 5.89 ± 2.48 J for SST, respectively). The value of the adhesiveness (in both groups of cheeses) increased (in absolute value). Finally, the cohesiveness of the cheeses increased in both groups, showing their maximum value on the 180th day (0.36 ± 0.02 for TC and 0.35 ± 0.01 for SST). As an exception, at day 90, the cohesiveness of the TC cheese decreased. Overall, the values of the textural characteristics of both TC- and SST-ripened cheeses reported here are similar to those reported by Kaminarides et al. [33], with the exception of the adhesiveness and gumminess parameters.

### 3.7. Microbiological Evolution during Cheese Ripening–Preservation

The microbiological results, total viable count (TVC), and molds/yeasts concerning the cheeses of the two groups and the respective brines during their ripening–preservation are presented in Table 12 and Table 13.

The TVCs in both cheeses and brines decreased over the ripening–preservation period. The mesophilic flora had quite similar values in both the cheeses and brines of the two groups. No statistically significant differences (*p* > 0.05) were found between the cheeses of the two groups and the respective brines, except for the 90th day, when the brine at TC showed a significantly (*p* = 0.041) higher TVC than that of the SST brine.

Molds/yeasts counts were low in both TC- and SST-ripened cheeses and brines. Different ripening–preservation containers did not statistically significantly (*p* > 0.05) affect the population of molds/yeasts, neither in the cheeses nor in the brines. The results regarding the TVC and the molds/yeasts, observed in both TC and SST cheeses during their ripening–preservation, agree with those reported in other studies [22,34].

### 3.8. Sensory Evaluation

The results of the sensory evaluation of the cheeses are presented in the Table 14. Specifically, the TC- and SST-ripened cheeses did not show significant differences (*p* > 0.05) between them regarding their appearance/color, structure/texture, and flavor/odor, with their scores being similar throughout their ripening–preservation. On days 60 and 90, the cheeses (of both groups) had the characteristics of mature cheeses, with a semi-hard texture and a pleasant acidic taste. Both cheese groups obtained the highest total score on day 90 (91.45% ± 1.97 for TC and 90.95% ± 4.60 for SST, with coefficients of 4 and 5, respectively). The high score of the structure parameter, on day 90, was expected due to the fact that the gradual accumulation of total solids and large water-soluble peptides showed their maximum value for both groups of cheeses on day 90 as well. However, the cheeses of both groups received lower scores on day 180, which was expected since proteolysis and lipolysis are more intense during that time. The cheeses were characterized by a pleasant, acidic taste.

## 4. Conclusions

The results presented in this study showed that the material, and the capacity, of the ripening–preservation containers did not statistically significantly affect the physicochemical, textural, microbiological, and sensory characteristics of the white brine cheeses. It should be noted that those white brine cheeses that are ripened and preserved in tin containers, and those that are ripened and preserved in stainless steel tanks, did not differ from each other in the two main parameters—their fat and moisture content—that determine the quality of cheese, according to the Greek legislation. Although no significant differences were observed between the white cheeses in brine that ripened and preserved in tin containers and stainless steel tanks, we consider that the latter can be used by cheese factories where a considerable part of their production is planned for repackaging as a SST container has many advantages, such as its reusability, resistance to corrosion and low temperatures, contribution to the product’s hygiene and protection, and minimizing of losses during repackaging.

## Figures and Tables

**Figure 1 foods-12-02332-f001:**
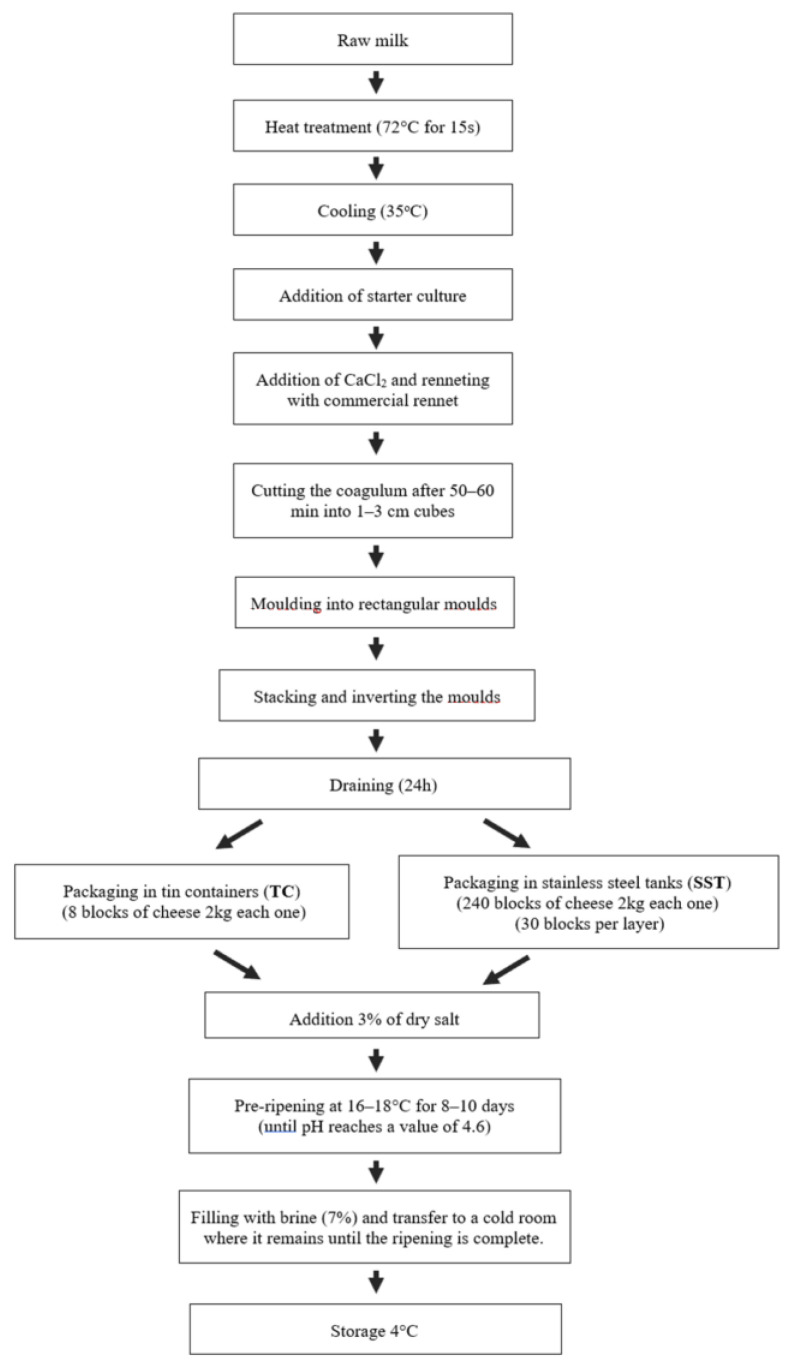
Flow chart for white brine cheeses (WBC) ripened and preserved in tin containers (TC) and stainless-steel tanks (SST).

**Figure 2 foods-12-02332-f002:**
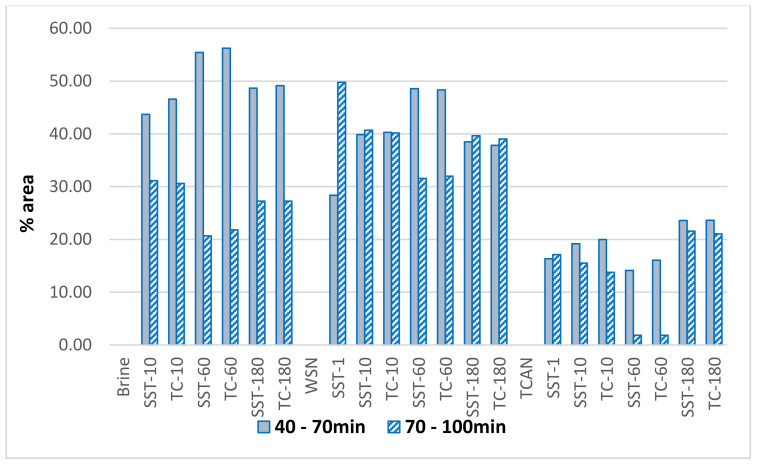
Evolution of characteristic regions of RP-HPLC profiles of brine and WSN/TN and TCA-N/TN of cheeses.

**Table 1 foods-12-02332-t001:** Physico-chemical composition (%) and pH of cheeses during ripening–preservation in tin containers (TC) and stainless steel tanks (SST) (Means ± S.D.).

Composition		Ripening Time (Days)							
	1 *	10		60		90		180	
		TC	SST	TC	SST	TC	SST	TC	SST
Moisture	56.29 ± 1.30	55.13 ± 1.53 ^a^	52.96 ± 1.00 ^a^	53.04 ± 1.39 ^a^	54.32 ± 0.57 ^a^	56.34 ± 2.00 ^a^	54.13 ± 1.68 ^a^	54.89 ± 1.84 ^a^	54.63 ± 1.24 ^a^
Fat	22.79 ± 1.86	23.23 ± 0.62 ^a^	23.47 ± 0.92 ^a^	24.25 ± 1.21 ^a^	23.84 ± 0.87 ^a^	23.36 ± 1.83 ^a^	24.58 ± 0.40 ^a^	23.31 ± 1.67 ^a^	24.38 ± 1.90 ^a^
Fat (in dry matter)	52.15 ± 4.25	51.77 ± 1.37 ^a^	49.90 ± 1.96 ^a^	51.64 ± 2.59 ^a^	52.18 ± 1.91 ^a^	53.50 ± 4.20 ^a^	53.59 ± 0.87 ^a^	51.67 ± 3.70 ^a^	53.44 ± 4.20 ^a^
Protein	16.23 ± 1.46	15.76 ± 0.63 ^a^	16.15 ± 0.20 ^a^	16.09 ± 0.35 ^a^	15.90 ± 0.19 ^a^	15.65 ± 1.09 ^a^	16.62 ± 0.59 ^a^	15.93 ± 1.17 ^a^	15.86 ± 0.58 ^a^
Protein (in dry matter)	37.14 ± 3.34	35.12 ± 1.40 ^a^	34.34 ± 0.42 ^a^	34.26 ± 0.75 ^a^	34.80 ± 0.41 ^a^	35.84 ± 2.49 ^a^	36.24 ± 1.29 ^a^	35.32 ± 2.59 ^a^	34.95 ± 1.28 ^a^
Ash	1.45 ± 0.06	3.30 ± 0.34 ^a^	3.38 ± 0.36 ^a^	3.29 ± 0.23 ^a^	3.44 ± 0.32 ^a^	3.37 ± 0.15 ^a^	3.56 ± 0.31 ^a^	3.74 ± 0.17 ^a^	3.81 ± 0.19 ^a^
Salt	0.23 ± 0.03	2.56 ± 0.43 ^a^	2.74 ± 0.62 ^a^	2.77 ± 0.29 ^a^	2.94 ± 0.30 ^a^	2.79 ± 0.02 ^a^	2.90 ± 0.24 ^a^	3.19 ± 0.21 ^a^	3.24 ± 0.39 ^a^
pH	4.72 ± 0.04	4.43 ± 0.12 ^a^	4.45 ± 0.08 ^a^	4.40 ± 0.10 ^a^	4.46 ± 0.15 ^a^	4.35 ± 0.12 ^a^	4.43 ± 0.16 ^a^	4.44 ± 0.06 ^a^	4.42 ± 0.05 ^a^

^a^ Means in each parameter, with the same superscripts for TC and SST values, do not differ statistically significantly (*p* > 0.05) in each time interval. * Before packaging.

**Table 2 foods-12-02332-t002:** pH, proteins, and salt concentration of brine (%) during ripening–preservation in tin containers (TC) and stainless steel tanks (SST). (Means ± S.D.).

Composition	Ripening Time (Days)							
	10		60		90		180	
	TC	SST	TC	SST	TC	SST	TC	SST
pH	4.53 ± 0.14 ^a^	4.49 ± 0.15 ^a^	4.57 ± 0.07 ^a^	4.59 ± 0.09 ^a^	4.40 ± 0.16 ^a^	4.48 ± 0.12 ^a^	4.31 ± 0.08 ^a^	4.35 ± 0.05 ^a^
Protein	1.36 ± 0.17 ^a^	0.95 ± 0.30 ^a^	2.21 ± 0.58 ^a^	1.91 ± 0.30 ^a^	2.72 ± 0.53 ^a^	2.43 ± 0.52 ^a^	2.40 ± 0.71 ^a^	2.23 ± 0.65 ^a^
Salt	4.85 ± 1.08 ^a^	5.67 ± 0.55 ^a^	4.97 ± 0.98 ^a^	4.63 ± 0.52 ^a^	4.36 ± 0.19 ^a^	4.56 ± 0.59 ^a^	5.10 ± 0.52 ^a^	5.09 ± 0.65 ^a^

^a^ Means in each parameter, with the same superscripts for TC and SST values, do not differ significantly (*p* > 0.05) in each time interval.

**Table 3 foods-12-02332-t003:** Major mineral concentration (mg/100 g) of cheeses during ripening and preservation in tin containers (TC) and stainless steel tanks (SST). (Means ± S.D.).

Mineral		Ripening Time (Days)							
	1 *	10		60		90		180	
		TC	SST	TC	SST	TC	SST	TC	SST
Calcium	358.04 ± 78.03	270.51 ± 37.39 ^a^	242.27 ± 80.76 ^a^	267.70 ± 42.22 ^a^	243.94 ± 25.43 ^a^	313.65 ± 44.41 ^a^	280.59 ± 63.60 ^a^	223.96 ± 56.13 ^a^	235.05 ± 62.78 ^a^
Magnesium	23.26 ± 4.11	19.43 ± 1.93 ^a^	17.60 ± 3.60 ^a^	17.40 ± 1.53 ^a^	16.58 ± 1.01 ^a^	19.39 ± 1.75 ^a^	17.23 ± 3.16 ^a^	14.62 ± 2.79 ^a^	14.30 ± 1.73 ^a^
Potassium	77.36 ± 12.73	63.11 ± 1.49 ^a^	53.11 ± 10.19 ^a^	62.29 ± 4.66 ^a^	59.42 ± 5.23 ^a^	86.95 ± 8.57 ^a^	75.72 ± 8.65 ^a^	48.65 ± 18.44 ^a^	59.79 ± 16.92 ^a^
Sodium	148.13 ± 88.97	1241.20 ± 424.97 ^a^	1188.02 ± 283.23 ^a^	1053.40 ± 129.96 ^a^	1201.44 ± 167.23 ^a^	1084.63 ± 54.49 ^a^	1167.88 ± 137.49 ^a^	1215.50 ± 77.37 ^a^	1266.06 ± 125.15 ^a^

^a^ Means in each parameter with the same superscripts for TC and SST values do not differ statistically significantly (*p* > 0.05) in each time interval. * Before packaging.

**Table 4 foods-12-02332-t004:** Major mineral concentration (mg/100g) of brines during ripening and preservation in tin containers (TC) and stainless steel tanks (SST). (Means ± S.D.).

Mineral	Ripening Time (Days)							
	10		60		90		180	
	TC	SST	TC	SST	TC	SST	TC	SST
Calcium	432.58 ± 19.34 ^a^	290.67 ± 81.27 ^b^	415.74 ± 93.71 ^a^	363.99 ± 94.00 ^a^	450.60 ± 29.00 ^a^	413.09 ± 74.07 ^a^	372.33 ± 52.00 ^a^	359.97 ± 28.60 ^a^
Magnesium	26.24 ± 1.42 ^a^	21.47 ± 4.01 ^a^	28.18 ± 3.91 ^a^	24.60 ± 3.28 ^a^	29.91 ± 1.47 ^a^	28.42 ± 4.04 ^a^	27.85 ± 3.57 ^a^	25.92 ± 1.99 ^a^
Potassium	98.09 ± 4.65 ^a^	82.68 ± 10.67 ^a^	104.80 ± 22.94 ^a^	92.73 ± 10.86 ^a^	108.54 ± 0.19 ^a^	108.46 ± 11.64 ^a^	120.09 ± 22.39 ^a^	130.12 ± 4.05 ^a^
Sodium	2421.42 ± 539.25 ^a^	2797.89 ± 164.04 ^a^	2471.68 ± 407.09 ^a^	2371.95 ± 169.13 ^a^	2046.38 ± 57.98 ^a^	2224.34 ± 218.37 ^a^	2255.00 ± 171.84 ^a^	2234.93 ± 369.30 ^a^

^a,b^ Means in each parameter with different superscripts for TC and SST values differ statistically significantly (*p* > 0.05) in each time interval.

**Table 5 foods-12-02332-t005:** Nitrogenous fractions of WSN/TN, TCA-N/TN, and TCA-N/WSN of cheeses during ripening and preservation in tin containers (TC) and stainless steel tanks (SST). (Means ± S.D.).

Nitrogenous Fractions		Ripening Time (Days)							
	1 *	10		60		90		180	
		TC	SST	TC	SST	TC	SST	TC	SST
%WSN/TN	8.19 ± 088 ^b^	10.29 ± 2.51 ^a^	10.41 ± 2.46 ^a^	9.99 ±2.45 ^a^	9.48 ± 1.80 ^a^	11.69 ± 2.30 ^a^	10.67 ± 2.84 ^a^	10.21 ± 4.89 ^a^	9.99 ± 3.81 ^a^
%TCA-N/TN	3.84 ± 0.15 ^b^	7.31 ± 1.53 ^a^	6.33 ± 1.23 ^a^	6.82 ± 1.69 ^a^	6.00 ± 0.78 ^a^	10.03 ± 1.07 ^a^	8.05 ± 2.40 ^a^	9.30 ± 3.31 ^a^	8.97 ± 2.72 ^a^
%TCA-N/WSN	47.31 ± 6.34 ^b^	71.50 ± 6.19 ^a^	61.35 ± 5.28 ^a^	68.20 ± 2.71 ^a^	64.16 ± 8.34 ^a^	87.07 ± 9.92 ^a^	75.61 ± 9.05 ^a^	94.40 ± 11.43 ^a^	91.76 ± 11.56 ^a^

^a.b^ Means in each parameter with different superscripts for TC and SST values differ statistically significantly (*p* > 0.05) in each time interval. * Before packaging.

**Table 6 foods-12-02332-t006:** Peptide areas in RP-HPLC profiles of cheese brine during ripening and preservation in tin containers (TC) and stainless steel tanks (SST). (Means ± S.D.).

RP-HPLC Profiles	Ripening Time (Days)					
	10		60		180	
	TC	SST	TC	SST	TC	SST
0–10 min	9.71 ± 1.73 ^a^	10.65 ± 1.02 ^a^	9.63 ± 1.68 ^a^	10.63 ± 2.31 ^a^	7.06 ± 0.77 ^a^	6.88 ± 0.94 ^a^
10–40 min	9.89 ± 2.26 ^a^	10.85 ± 1.90 ^a^	11.96 ± 2.81 ^a^	12.75 ± 2.18 ^a^	13.31 ± 1.29 ^a^	13.45 ± 1.57 ^a^
40–70 min	46.60 ± 1.87 ^a^	43.68 ± 3.74 ^a^	56.25 ± 2.46 ^a^	55.41 ± 2.70 ^a^	49.14 ± 3.92 ^a^	48.65 ± 3.59 ^a^
70–100 min	30.62 ± 2.25 ^a^	31.17 ± 2.90 ^a^	21.81 ± 1.97 ^a^	20.69 ± 1.41 ^a^	27.27 ± 1.33 ^a^	27.25 ± 0.39 ^a^
HB/HL ^1^	1.45 ± 0.21 ^a^	1.51 ± 0.24 ^a^	0.96 ± 0.21 ^a^	0.90 ± 0.14 ^a^	1.07 ± 0.06 ^a^	1.08 ± 0.04 ^a^
HB/HL ^2^	0.46 ± 0.06 ^a^	0.48 ± 0.07 ^a^	0.28 ± 0.03 ^a^	0.26 ± 0.02 ^a^	0.39 ± 0.03 ^a^	0.40 ± 0.01 ^a^

^a^ Means in each parameter with the same superscripts for TC and SST values do not differ statistically significantly (*p* > 0.05) in each time interval. 1: Ratio of the area of peaks eluted from 55 to 100 min (hydrophobic peptides (HB)), to those eluted from 10 to 55 min (hydrophilic peptides (HL)). 2: Ratio of the area of peaks eluted from 70 to 100 min (hydrophobic peptides (HB)), to those eluted from 0 to 70 min (hydrophilic peptides (HL)).

**Table 7 foods-12-02332-t007:** Peptide areas in RP-HPLC profiles of the WSN/TN fraction of the cheeses during ripening and preservation in tin containers (TC) and stainless steel tanks (SST). (Means ± S.D.).

RP-HPLC Profiles	Ripening Time (Days)					
	10		60		180	
	TC	SST	TC	SST	TC	SST
0–10 min	5.18 ± 1.55 ^a^	4.91 ± 1.63 ^a^	8.05 ± 1.14 ^a^	8.43 ± 0.67 ^a^	3.85 ± 0.48 ^a^	3.97 ± 0.11 ^a^
10–40 min	7.79 ± 1.76 ^a^	7.88 ± 1.77 ^a^	10.60 ± 5.08 ^a^	10.50 ± 4.27 ^a^	10.85 ± 0.43 ^a^	10.84 ± 0.91 ^a^
40–70 min	40.31 ± 5.11 ^a^	39.84 ± 4.59 ^a^	48.33 ± 0.80 ^a^	48.56 ± 1.45 ^a^	37.81 ± 3.59 ^a^	38.49 ± 2.05 ^a^
70–100 min	40.20 ± 3.70 ^a^	40.73 ± 2.71 ^a^	32.01 ± 4.10 ^a^	31.56 ± 2.79 ^a^	39.05 ± 1.91 ^a^	39.63 ± 1.00 ^a^
HB/HL ^1^	2.06 ± 0.28 ^a^	2.23 ± 0.15 ^a^	1.24 ± 0.35 ^a^	1.23 ± 0.17 ^a^	1.75 ± 0.18 ^a^	1.79 ± 0.08 ^a^
HB/HL ^2^	0.76 ± 0.13 ^a^	0.78 ± 0.11 ^a^	0.48 ± 0.09 ^a^	0.47 ± 0.06 ^a^	0.75 ± 0.09 ^a^	0.74 ± 0.03 ^a^

^a^ Means in each parameter with the same superscripts for TC and SST values do not differ statistically significantly (*p* > 0.05) in each time interval. 1: Ratio of the area of peaks eluted from 55 to 100 min (hydrophobic peptides (HB)), to those eluted from 10 to 55 min (hydrophilic peptides (HL)). 2: Ratio of the area of peaks eluted from 70 to 100 min (hydrophobic peptides (HB)), to those eluted from 0 to 70 min (hydrophilic peptides (HL)).

**Table 8 foods-12-02332-t008:** Peptide areas in RP-HPLC profiles of the TCA-N/TN fraction of the cheeses during ripening and preservation in tin containers (TC) and stainless steel tanks (SST). (Means ± S.D.).

RP-HPLC Profiles	Ripening Time (Days)					
	10		60		180	
	TC	SST	TC	SST	TC	SST
0–10 min	52.16 ± 1.00 ^a^	51.27 ± 0.73 ^a^	72.31 ± 7.14 ^a^	74.51 ± 8.40 ^a^	40.40 ± 1.41 ^a^	39.96 ± 0.60 ^a^
10–40 min	7.40 ± 0.55 ^a^	7.18 ± 0.18 ^a^	9.21 ± 2.83 ^a^	9.07 ± 3.39 ^a^	6.95 ± 0.26 ^a^	6.85 ± 0.16 ^a^
40–70 min	20.00 ± 1.05 ^a^	19.16 ± 1.02 ^a^	16.07 ± 4.01 ^a^	14.12 ± 3.73 ^a^	23.61 ± 1.94 ^a^	23.59 ± 2.32 ^a^
70–100 min	13.79 ± 1.07 ^a^	15.55 ± 1.24 ^a^	1.87 ± 1.36 ^a^	1.86 ± 1.63 ^a^	21.07 ± 1.09 ^a^	21.60 ± 1.84 ^a^
HB/HL ^1^	1.46 ± 0.08 ^a^	1.66 ± 0.17 ^a^	0.45 ± 0.06 ^a^	0.45 ± 0.05 ^a^	1.71 ± 0.17 ^a^	1.83 ± 0.23 ^a^
HB/HL ^2^	0.17 ± 0.01 ^a^	0.20 ± 0.02 ^a^	0.02 ± 0.01 ^a^	0.02 ± 0.02 ^a^	0.30 ± 0.02 ^a^	0.31 ± 0.04 ^a^

^a^ Means in each parameter with the same superscripts for TC and SST values do not differ statistically significantly (*p* > 0.05) in each time interval. 1: Ratio of the area of peaks eluted from 55 to 100 min (hydrophobic peptides (HB)), to those eluted from 10 to 55 min (hydrophilic peptides (HL)). 2: Ratio of the area of peaks eluted from 70 to 100 min (hydrophobic peptides (HB)), to those eluted from 0 to 70 min (hydrophilic peptides (HL)).

**Table 9 foods-12-02332-t009:** Means ± S.D of fatty acid composition (g/100g) of cheeses during ripening and preservation in tin containers (TC) and stainless steel tanks (SST).

Fatty Acids	Ripening Time (Days)					
	60		90		180	
	TC	SST	TC	SST	TC	SST
C4	3.36 ± 0.90 ^a^	2.59 ± 0.30 ^a^	2.56 ± 0.30 ^a^	4.26 ± 2.52 ^a^	2.78 ± 0.32 ^a^	2.93 ± 0.30 ^a^
C6	1.98 ± 0.56 ^a^	1.83 ± 0.16 ^a^	1.87 ± 0.43 ^a^	2.12 ± 0.50 ^a^	2.01 ± 0.32 ^a^	2.10 ± 0.33 ^a^
C8	2.97 ± 0.99 ^a^	2.27 ± 0.13 ^a^	2.41 ± 0.19 ^a^	2.76 ± 0.06 ^b^	2.77 ± 0.15 ^a^	2.96 ± 0.08 ^a^
C10	10.28 ± 3.56 ^a^	8.35 ± 0.20 ^a^	8.59 ± 0.45 ^a^	9.05 ± 0.52 ^a^	9.34 ± 0.49 ^a^	9.77 ± 0.12 ^a^
C12	5.05 ± 0.91 ^a^	4.78 ± 0.31 ^a^	4.72 ± 0.24 ^a^	5.03 ± 0.35 ^a^	5.24 ± 0.25 ^a^	5.42 ± 0.22 ^a^
C14	13.84 ± 0.14 ^a^	13.34 ± 0.22 ^b^	13.42 ± 0.15 ^a^	13.78 ± 0.05 ^b^	14.21 ± 0.38 ^a^	14.01 ± 0.26 ^a^
C15	0.82 ± 0.14 ^a^	1.77 ± 1.35 ^a^	0.83 ± 0.13 ^a^	0.93 ± 0.12 ^a^	1.05 ± 0.03 ^a^	1.01 ± 0.06 ^a^
C16	30.87 ± 2.08 ^a^	30.72 ± 0.65 ^a^	31.14 ± 0.83 ^a^	31.08 ± 1.87 ^a^	30.67 ± 0.64 ^a^	30.24 ± 0.37 ^a^
C17	0.40 ± 0.02 ^a^	0.39 ± 0.12 ^a^	0.41 ± 0.06 ^a^	0.50 ± 0.25 ^a^	0.48 ± 0.03 ^a^	0.46 ± 0.03 ^a^
C18	7.69 ± 1.04 ^a^	8.24 ± 0.33 ^a^	8.55 ± 0.62 ^a^	8.20 ± 0.67 ^a^	7.67 ± 0.37 ^a^	7.51 ± 0.24 ^a^
C14:1	0.39 ± 0.09 ^a^	0.33 ± 0.13 ^a^	0.35 ± 0.06 ^a^	0.38 ± 0.05 ^a^	0.44 ± 0.07 ^a^	0.46 ± 0.03 ^a^
C16:1	1.71 ± 0.26 ^a^	1.54 ± 0.41 ^a^	1.36 ± 0.23 ^a^	1.79 ± 0.43 ^a^	1.52 ± 0.06 ^a^	1.49 ± 0.05 ^a^
C18:1 n9	19.14 ± 2.48 ^a^	23.84 ± 5.08 ^a^	20.72 ± 1.10 ^a^	19.66 ± 1.07 ^a^	19.42 ± 0.63 ^a^	19.13 ± 0.54 ^a^
C18:2 n6 t	1.62 ± 0.39 ^a^	1.88 ± 0.21 ^a^	1.81 ± 0.13 ^a^	1.77 ± 0.15 ^a^	1.91 ± 0.10 ^a^	1.87 ± 0.12 ^a^
C18:2 n6 c	0.27 ± 0.10 ^a^	0.31 ± 0.08 ^a^	0.34 ± 0.04 ^a^	0.31 ± 0.04 ^a^	0.36 ± 0.02 ^a^	0.35 ± 0.02 ^a^
C18:3 n3	0.04 ± 0.08 ^a^	0.10 ± 0.03 ^a^	0.09 ± 0.08 ^a^	0.13 ± 0.17 ^a^	0.15 ± 0.02 ^a^	0.08 ± 0.07 ^a^
CLA	0.45 ± 0.32 ^a^	1.76 ± 1.21 ^a^	1.29 ± 1.11 ^a^	2.02 ± 1.36 ^a^	0.50 ± 0.03 ^a^	0.63 ± 0.44 ^a^
SFA	77.26 ± 3.94 ^a^	74.28 ± 0.77 ^a^	74.49 ± 0.44 ^a^	77.71 ± 5.20 ^a^	76.21 ± 0.64 ^a^	76.41 ± 0.54 ^a^
MUFA	21.25 ± 2.26 ^a^	25.70 ± 5.27 ^a^	22.43 ± 0.82 ^a^	21.82 ± 1.06 ^a^	21.39 ± 0.61 ^a^	21.08 ± 0.51 ^a^
PUFA	2.38 ± 0.82 ^a^	4.05 ± 0.94 ^a^	3.52 ± 0.99 ^a^	4.24 ± 1.26 ^a^	2.91 ± 0.15 ^a^	2.92 ± 0.43 ^a^

^a,b^ Means in each parameter with different superscripts for TC and SST values differ statistically significantly (*p* > 0.05) in each time interval.

**Table 10 foods-12-02332-t010:** Means ± S.D of volatile compounds (mg/kg) identified in cheeses during ripening and preservation in tin containers (TC) and stainless steel tanks (SST).

Volatiles Compounds	Ripening Time (Days)					
	60	90	180	60	90	180
	TC			SST		
**Organic Acids (15)**						
9-Decenoic acid	0.36 ± 0.62	18.65 ± 11.17	38.16 ± 10.33	31.85 ± 7.89	26.62 ± 8.62	48.49 ± 6.41
Acetic acid	281.13 ± 35.88	354.16 ± 70.56	653.86 ± 100.41	250.82 ± 180.86	251.14 ± 89.48	726.87 ± 141.69
Benzoic acid	16.58 ± 13.47	58.28 ± 23.05	62.76 ± 23.37	57.54 ± 28.03	55.34 ± 12.06	112.70 ± 42.80
Butanoic acid	132.32 ± 64.25	424.67 ± 47.91	337.25 ± 93.49	217.51 ± 121.15	521.14 ± 129.51	352.88 ± 142.39
Decanoic acid	224.62 ± 69.49	621.97 ± 32.47	782.20 ± 124.09	199.63 ± 171.66	676.52 ± 154.63	708.49 ± 165.78
Dodecanoic acid	23.02 ± 16.72	83.74 ± 19.74	130.33 ± 89.16	214.73 ± 45.25	144.07 ± 77.19	250.67 ± 56.09
Heptanoic acid	2.68 ± 2.92	52.98 ± 22.32	12.29 ± 4.11	8.89 ± 9.22	17.74 ± 6.84	25.74 ± 11.79
Hexanoic acid	276.36 ± 28.18	660.92 ± 136.16	706.56 ± 63.05	269.36 ± 120.58	737.93 ± 148.76	808.06 ± 142.36
Isovaleric acid	0.00 ± 0.00	12.12 ± 10.25	1.54 ± 1.07	1.03 ± 1.47	3.05 ± 1.60	1.13 ± 0.88
Nonanoic acid	4.12 ± 2.68	14.96 ± 6.94	17.38 ± 9.23	27.38 ± 13.04	22.95 ± 5.67	37.33 ± 10.19
Octanoic acid	251.58 ± 41.99	925.58 ± 69.82	1013.75 ± 181.24	365.49 ± 119.99	744.87 ± 137.96	655.13 ± 160.15
Pentanoic acid	1.22 ± 1.04	7.71 ± 11.36	6.46 ± 3.37	10.01 ± 4.38	8.91 ± 4.81	12.32 ± 4.90
Propanoic acid	1.20 ± 2.08	4.79 ± 6.96	12.16 ± 5.16	35.49 ± 3.74	2.67 ± 8.21	26.75 ± 11.62
Tetradecanoic acid	1.38 ± 0.39	20.63 ± 13.19	42.10 ± 9.73	63.51 ± 7.38	2.98 ± 36.20	71.59 ± 7.56
Undecanoic acid	0.00 ± 0.00	4.35 ± 3.52	11.49 ± 9.43	11.56 ± 9.00	6.77 ± 108.10	7.87 ± 2.21
**Total acids**	**1216.57**	**3265.51**	**3828.29**	**1764.80**	**3222.70**	**3846.02**
**Alcohols (16)**						
Ethanol	257.21 ± 28.63	371.16 ± 37.90	420.53 ± 77.87	260.03 ± 48.85	377.96 ± 54.26	406.17 ± 133.74
1-Propanol	6.06 ± 2.85	18.00 ± 10.08	47.12 ± 22.04	15.18 ± 34.20	22.99 ± 1.92	43.63 ± 29.57
2-Methyl-1-propanol	10.29 ± 3.59	10.05 ± 3.32	8.77 ± 2.47	8.96 ± 2.48	8.95 ± 2.65	8.40 ± 2.40
2-Propen-1-ol	0.51 ± 0.17	4.27 ± 2.34	5.02 ± 3.46	5.72 ± 2.79	2.87 ± 2.34	6.81 ± 3.60
1-Pentanol	10.43 ± 2.14	13.19 ± 4.87	3.29 ± 11.04	6.70 ± 13.79	4.56 ± 11.60	4.49 ± 11.13
1-Hexanol	4.22 ± 4.04	10.71 ± 14.36	9.46 ± 6.37	13.01 ± 7.38	11.91 ± 7.81	15.32 ± 7.90
1-Heptanol	0.43 ± 0.14	3.19 ± 2.87	1.29 ± 1.04	4.70 ± 3.79	2.56 ± 1.60	2.49 ± 1.13
1-Octen-3-ol	0.28 ± 0.17	0.74 ± 0.63	0.89 ± 0.25	1.67 ± 0.79	1.48 ± 0.74	2.90 ± 1.54
2-Nonanol	0.00 ± 0.00	1.80 ± 1.15	0.75 ± 0.46	3.50 ± 2.81	2.11 ± 1.22	1.91 ± 1.67
Decanol	2.29 ± 1.59	2.05 ± 1.32	0.77 ± 0.47	0.96 ± 0.48	0.95 ± 0.65	0.40 ± 0.40
Dodecanol	0.59 ± 0.41	0.67 ± 0.65	2.49 ± 1.51	5.52 ± 5.80	1.49 ± 1.28	9.85 ± 5.65
Tridecanol	0.18 ± 0.11	0.81 ± 0.53	2.59 ± 1.06	3.95 ± 4.18	2.14 ± 2.69	3.94 ± 2.42
Tetradecanol	0.00 ± 0.00	1.58 ± 1.30	1.61 ± 1.31	2.10 ± 1.72	0.87 ± 0.66	4.37 ± 3.70
Hexadecanol	0.31 ± 0.18	1.38 ± 1.62	1.77 ± 1.25	0.64 ± 0.26	0.94 ± 0.24	2.68 ± 1.97
Benzene ethanol	12.76 ± 4.40	95.47 ± 46.65	21.58 ± 4.31	19.99 ± 11.25	12.73 ± 5.88	32.29 ± 17.39
2-Phenyl ethanol	0.43 ± 0.17	1.75 ± 1.01	2.48 ± 1.76	1.73 ± 1.41	2.35 ± 3.42	3.79 ± 3.94
**Total Alcohol**	**305.99**	**536.82**	**530.41**	**354.36**	**456.86**	**549.44**
**Aldehydes (10)**						
Acetaldehyde	1.82 ± 0.52	1.28 ± 0.33	1.13 ± 0.65	1.92 ± 1.02	0.44 ± 0.66	0.96 ± 0.54
Furfural	0.02 ± 0.10	0.25 ± 0.10	0.82 ± 0.22	0.52 ± 0.14	0.22 ± 0.25	0.61 ± 0.25
Hexanal	3.83 ± 2.20	3.59 ± 2.50	2.31 ± 1.45	2.50 ± 1.02	2.49 ± 1.35	1.94 ± 2.02
Heptanal	0.00 ± 0.00	0.60 ± 0.54	1.44 ± 1.08	0.65 ± 0.81	0.70 ± 0.26	1.49 ± 1.99
Nonanal	1.69 ± 0.59	2.68 ± 1.30	3.87 ± 3.82	1.80 ± 1.66	3.36 ± 3.58	3.91 ± 3.54
Decanal	1.20 ± 1.05	1.82 ± 1.43	1.57 ± 1.13	2.39 ± 1.02	2.52 ± 2.07	3.55 ± 2.44
2,4-Dimethylpentanal	0.89 ± 0.25	1.96 ± 1.35	2.35 ± 1.20	1.22 ± 1.20	1.52 ± 1.32	3.26 ± 1.88
3-Hydroxybutanal	1.97 ± 1.25	1.70 ± 1.84	0.85 ± 0.55	0.86 ± 0.22	1.03 ± 1.20	0.78 ± 0.33
2-methylpentanal	0.79 ± 0.55	1.03 ± 0.88	1.89 ± 1.33	1.36 ± 1.25	1.66 ± 1.35	6.03 ± 3.25
Benzaldehyde	0.62 ± 0.23	1.63 ± 1.46	3.55 ± 2.28	2.71 ± 2.35	1.54 ± 3.37	5.27 ± 2.54
**Total Aldehydes**	**12.83**	**16.54**	**19.78**	**15.93**	**15.48**	**27.80**
**Esters (13)**						
2-Hydroxy-propanoic acid. ethyl ester	1.52 ± 0.64	2.56 ± 1.56	1.32 ± 0.99	0.03 ± 0.03	3.80 ± 2.29	8.64 ± 4.40
Aceticacid. 2-phenylethyl ester	1.86 ± 1.03	86.44 ± 24.37	60.60 ± 10.04	26.77 ± 15.73	15.91 ± 6.88	35.22 ± 15.46
Citronellylformate	0.92 ± 0.37	2.94 ± 1.01	0.76 ± 0.66	4.15 ± 2.77	1.81 ± 0.97	2.49 ± 2.13
Decanoic acid. Ethylester	9.29 ± 1.95	37.54 ± 14.44	37.84 ± 11.69	28.22 ± 19.96	39.01 ± 24.71	115.30 ± 60.57
Dihydrocitronellol acetate	0.42 ± 0.16	0.35 ± 0.16	1.62 ± 0.54	0.94 ± 0.26	1.19 ± 0.77	0.51 ± 0.13
Ethyl acetate	1.00 ± 0.73	11.22 ± 11.41	19.59 ± 16.54	7.76 ± 10.04	1.52 ± 1.21	18.48 ± 15.57
Hexadecanoic acid. ethylester	0.55 ± 0.48	1.09 ± 0.96	2.45 ± 1.30	1.71 ± 3.57	1.70 ± 1.88	2.86 ± 3.87
Hexanoic acid. ethylester	1.77 ± 2.81	3.09 ± 1.69	3.99 ± 2.80	3.94 ± 2.56	3.77 ± 2.81	8.73 ± 3.81
Isopentyl formate	3.66 ± 1.34	0.50 ± 0.20	0.86 ± 0.25	0.55 ± 1.05	1.98 ± 2.52	0.95 ± 0.53
Isopentyl-isovalerate	0.30 ± 0.13	0.79 ± 0.23	2.07 ± 1.83	2.81 ± 1.31	2.17 ± 2.13	5.48 ± 4.04
Octanoic acid. ethylester	4.85 ± 2.93	6.45 ± 3.64	7.54 ± 4.11	9.27 ± 5.31	8.36 ± 6.71	34.37 ± 18.52
Tetradecanoi acid. ethylester	2.15 ± 0.97	3.54 ± 2.49	3.77 ± 3.09	6.28 ± 8.87	1.16 ± 1.34	9.36 ± 5.65
2-Ethenyloxy ethanol	0.75 ± 0.29	4.27 ± 2.09	77.59 ± 35.09	6.98 ± 4.52	20.30 ± 8.44	33.09 ± 67.68
**Total Esters**	**29.04**	**160.78**	**220.00**	**99.41**	**102.68**	**275.48**
**Ketones (11)**						
2-Butanone	4.90 ± 2.85	5.79 ± 2.50	7.04 ± 3.59	7.72 ± 4.25	7.51 ± 2.55	9.09 ± 3.55
2-Heptanone	2.68 ± 1.23	3.02 ± 1.85	3.76 ± 2.05	4.49 ± 3.20	3.26 ± 1.25	3.61 ± 1.64
2-Octanone	3.43 ± 2.22	4.24 ± 2.89	5.05 ± 3.66	6.34 ± 3.65	5.59 ± 2.38	6.63 ± 2.22
2-Nonanone	5.72 ± 3.25	7.77 ± 2.01	1.60 ± 0.22	1.34 ± 1.02	1.74 ± 0.80	1.04 ± 0.55
2-Decanone	2.30 ± 1.55	3.19 ± 2.99	4.44 ± 1.85	5.12 ± 3.81	4.91 ± 2.46	6.49 ± 5.02
2-Undecanone	0.08 ± 0.04	0.42 ± 0.37	1.16 ± 1.13	1.89 ± 1.79	0.66 ± 1.06	1.01 ± 1.67
2-Dodecanone	0.83 ± 0.31	1.64 ± 1.10	2.45 ± 2.83	3.74 ± 2.79	2.99 ± 1.19	4.03 ± 2.60
5-Methyl-2-Hexanone	5.12 ± 3.54	5.17 ± 2.95	2.75 ± 0.84	5.93 ± 2.23	2.19 ± 1.14	7.45 ± 4.25
5-Methyl-3-Heptanone	2.28 ± 1.95	6.78 ± 2.44	8.38 ± 5.12	12.76 ± 6.89	8.71 ± 4.88	19.46 ± 10.17
2-Methyl-4-heptanone	3.41 ± 1.99	7.71 ± 2.81	8.77 ± 10.99	14.84 ± 10.08	8.31 ± 6.71	19.46 ± 13.17
2-Piperidinone	0.14 ± 0.14	1.57 ± 1.27	1.35 ± 1.41	0.96 ± 0.22	1.06 ± 1.63	2.32 ± 2.00
**Total ketones**	**35.41**	**54.22**	**50.25**	**67.75**	**47.73**	**95.96**
**Lactones (5)**						
Gamma Nonalactone	1.02 ± 0.55	0.25 ± 0.22	1.28 ± 0.69	2.55 ± 0.65	2.05 ± 1.55	0.88 ± 0.33
Gamma-decalactone	0.37 ± 0.33	0.95 ± 0.28	1.83 ± 1.27	1.58 ± 1.00	0.91 ± 0.65	1.02 ± 1.07
Gamma-dodecalactone	4.52 ± 3.82	4.67 ± 2.04	7.36 ± 4.52	10.16 ± 6.28	4.77 ± 4.82	20.77 ± 13.87
Delta-nonalactone	2.95 ± 1.52	3.03 ± 1.48	4.00 ± 2.46	12.93 ± 9.83	5.51 ± 4.56	19.46 ± 13.17
Delta-decalactone	1.22 ± 0.80	2.05 ± 1.02	2.02 ± 1.08	0.58 ± 0.33	0.95 ± 0.88	1.64 ± 0.45
**Total Lactones**	**10.08**	**10.95**	**16.49**	**27.8** **0**	**14.19**	**43.77**
**Terpens (6)**						
Dehydro-apofarnesol	0.30 ± 0.22	1.48 ± 0.83	0.75 ± 0.63	2.44 ± 1.13	2.72 ± 2.58	3.82 ± 3.17
Dihydrocitronellol	0.51 ± 0.47	4.27 ± 2.34	5.02 ± 3.46	2.55 ± 1.25	2.87 ± 2.34	3.88 ± 1.05
Farnesol	0.43 ± 0.24	3.19 ± 2.87	1.29 ± 1.04	4.70 ± 3.79	2.56 ± 1.60	2.49 ± 2.13
Sesquilavandulol	0.00 ± 0.00	1.80 ± 0.95	0.75 ± 0.16	3.50 ± 2.81	2.11 ± 1.22	1.91 ± 1.67
Tetrahydro-lavandulol	0.43 ± 0.17	1.75 ± 0.81	2.48 ± 0.76	1.73 ± 1.41	2.35 ± 1.42	3.79 ± 2.94
Tetrahydro-citronellene	0.46 ± 0.19	1.06 ± 0.27	1.47 ± 1.46	0.97 ± 0.81	0.77 ± 0.17	3.56 ± 2.84
**Total Terpens**	**2.13**	**13.55**	**11.76**	**15.89**	**13.38**	**19.45**
**Alcanes (8)**						
2.2-Dimethylbutane	0.30 ± 0.26	0.85 ± 0.59	1.01 ± 0.45	0.76 ± 1.18	0.41 ± 0.19	0.62 ± 0.10
3-Methyl-hexane	0.56 ± 0.49	2.11 ± 1.43	0.83 ± 0.88	1.89 ± 0.96	0.41 ± 0.14	2.94 ± 3.39
Decane	1.46 ± 1.17	1.56 ± 0.69	11.75 ± 3.61	1.39 ± 4.79	2.95 ± 1.83	3.20 ± 1.20
Dodecane	0.92 ± 0.43	1.04 ± 1.29	3.02 ± 0.46	2.72 ± 4.15	1.61 ± 1.09	2.54 ± 1.41
Nonane	0.91 ± 0.88	1.33 ± 1.76	1.77 ± 1.01	2.92 ± 1.46	2.20 ± 1.08	2.90 ± 2.70
Tetradecane	0.54 ± 0.73	0.61 ± 0.15	0.81 ± 0.22	1.22 ± 0.59	0.80 ± 0.44	3.40 ± 2.10
Tridecane	1.00 ± 0.52	0.84 ± 1.53	1.42 ± 1.12	1.79 ± 1.89	0.85 ± 0.30	1.40 ± 2.47
Undecane	1.89 ± 1.14	4.11 ± 2.43	1.80 ± 0.76	2.60 ± 0.62	1.63 ± 1.10	1.39 ± 1.98
**Total Alcanes**	**7.58**	**12.45**	**22.41**	**15.29**	**10.86**	**18.39**
**Amines (3)**						
Amide	0.12 ± 0.15	0.05 ± 0.06	1.01 ± 1.02	1.06 ± 0.02	1.11 ± 1.02	0.32 ± 0.16
Piperidini	0.00 ± 0.00	0.16 ± 0.14	1.32 ± 1.24	0.02 ± 0.03	0.01 ± 0.02	1.24 ± 1.13
Dimethyl-amine	0.06 ± 0.03	0.11 ± 0.16	0.23 ± 0.12	0.03 ± 0.12	0.41 ± 0.55	0.24 ± 0.11
Total Amines	**0.182**	**0.326**	**2.564**	**1.093**	**1.536**	**1.806**
**Other aroma compounds (6)**						
2 H-Pyran-2-one tetrahydro-6-pentyl	2.28 ± 1.95	1.78 ± 0.44	0.38 ± 0.12	1.76 ± 1.89	1.71 ± 1.88	1.46 ± 1.17
2 H-Pyran-2-one tetrahydro-6-propyl	0.41 ± 1.99	0.71 ± 0.81	0.77 ± 1.99	1.84 ± 1.08	2.31 ± 0.71	1.46 ± 1.17
Phenol	0.28 ± 0.17	0.74 ± 0.63	0.89 ± 0.25	1.67 ± 0.79	1.48 ± 0.74	2.90 ± 1.54
Camphor	0.83 ± 0.58	1.83 ± 0.67	0.68 ± 4.34	1.97 ± 5.98	1.01 ± 4.13	1.63 ± 0.19
2 H-Pyran-2-one tetrahydro-6-pentyl	2.28 ± 1.95	0.78 ± 0.44	0.38 ± 5.12	1.76 ± 6.89	2.71 ± 2.88	1.46 ± 10.17
Styrene	2.62 ± 1.42	0.95 ± 0.54	1.79 ± 1.90	0.90 ± 0.63	0.82 ± 0.24	1.75 ± 8.51
**Total other aroma compounds**	**8.70**	**6.79**	**4.89**	**9.90**	**10.04**	**10.66**

**Table 11 foods-12-02332-t011:** Means ± S.D. of textural characteristics (hardness (N), adhesiveness (J), elasticity (mm), cohesiveness, gumminess (N), and chewiness (J)) of cheeses during ripening and preservation in tin containers (TC) and stainless steel tanks (SST).

Textural Parameters	Ripening Time (Days)							
	10		60		90		180	
	TC	SST	TC	SST	TC	SST	TC	SST
Hardness	8.28 ±2.05 ^a^	7.58 ± 0.91 ^a^	6.04 ± 1.90 ^a^	7.14 ± 2.47 ^a^	7.48 ± 2.58 ^a^	8.41 ± 1.40 ^a^	9.66 ± 1.10 ^a^	8.45 ± 2.29 ^a^
Adhesiveness	−31.38 ± 15.29 ^a^	−23.72 ± 6.41 ^a^	−22.50 ± 9.59 ^a^	−18.84 ± 12.90 ^a^	−24.95 ± 9.72 ^a^	−28.32 ± 8.12 ^a^	−33.39 ± 7.17 ^a^	−40.66 ± 13.86 ^a^
Elasticity	1.02 ± 0.02 ^a^	1.22 ± 0.36 ^a^	1.09 ± 0.09 ^a^	1.21 ± 0.20 ^a^	1.24 ± 0.34 ^a^	1.18 ± 0.30 ^a^	1.07 ± 0.06 ^a^	2.11 ± 1.14 ^a^
Cohesiveness	0.37 ± 0.03 ^a^	0.37 ± 0.07 ^a^	0.35 ± 0.03 ^a^	0.32 ± 0.06 ^a^	0.32 ± 0.04 ^a^	0.34 ± 0.04 ^a^	0.36 ± 0.02 ^a^	0.35 ± 0.01 ^a^
Gumminess	3.09 ± 0.92 ^a^	2.85 ± 0.88 ^a^	2.14 ± 0.83 ^a^	2.36 ± 1.16 ^a^	2.45 ± 1.07 ^a^	2.85 ± 0.70 ^a^	3.44 ± 0.37 ^a^	2.99 ± 0.86 ^a^
Chewiness	3.15 ± 0.93 ^a^	3.31 ± 0.47 ^a^	2.37 ± 1.07 ^a^	2.70 ± 0.86 ^a^	2.81 ± 0.67 ^a^	3.21 ± 0.13 ^a^	3.67 ± 0.26 ^a^	5.89 ± 2.48 ^a^

^a^ Means in each parameter with the same superscripts for TC and SST values do not differ statistically significantly (*p* > 0.05) in each time interval.

**Table 12 foods-12-02332-t012:** Microbial counts (log CFU/g) of cheeses for total viable count and molds/yeasts during ripening and preservation in tin containers (TC) and stainless steel tanks (SST). (Means ± S.D.).

Microbial Groups	Ripening Time (Days)								
	1 **	10		60		90		180	
		TC	SST	TC	SST	TC	SST	TC	SST
Total viable count	9.40 ± 0.25	8.75 ± 0.21 ^a^	8.45 ± 0.33 ^a^	7.56 ± 0.29 ^a^	6.62 ± 1.40 ^a^	7.39 ± 1.11 ^a^	5.92 ± 0.74 ^a^	6.50 ± 1.08 ^a^	5.47 ± 0.79 ^a^
Molds/Yeasts	1.63 ± 1.41 ^a^	3.44 ± 1.22 ^a^	3.16 ± 1.00 ^a^	3.24 ± 0.78 ^a^	3.45 ± 0.51 ^a^	2.34 ± 0.96 ^a^	3.18 ± 0.83 ^a^	1.87 ± 1.68 ^a^	2.67 ± 0.37 ^a^

^a^ Means in each parameter with the same superscripts for TC and SST values do not differ statistically significantly (*p* > 0.05) in each time interval. ** Before packaging.

**Table 13 foods-12-02332-t013:** Microbial counts (log cfu/g) of brines for total viable count and molds/yeasts during ripening and preservation in tin containers (TC) and stainless steel tanks (SST). (Means ± S.D.).

Microbial Groups	Ripening Time (Days)							
	10		60		90		180	
	TC	SST	TC	SST	TC	SST	TC	SST
Total viable count	7.13 ± 0.57 ^a^	7.13 ± 1.37 ^a^	7.37 ± 0.15 ^a^	6.90 ± 1.33 ^a^	7.51 ± 0.28 ^a^	6.19 ± 0.72 ^b^	6.81 ± 0.58 ^a^	5.29 ± 0.75 ^a^
Molds/Yeasts	4.35 ± 0.32 ^a^	3.51 ± 1.00 ^a^	4.18 ± 0.10 ^a^	4.77 ± 0.44 ^a^	3.44 ± 1.06 ^a^	4.15 ± 1.01 ^a^	3.17 ± 1.02 ^a^	3.83 ± 0.13 ^a^

^a,b^ Means in each parameter with different superscripts for TC and SST values differ statistically significantly (*p* > 0.05) in each time interval.

**Table 14 foods-12-02332-t014:** Sensory evaluation of cheeses during their ripening and preservation in tin containers (TC) and stainless steel tanks (SST). (Means ± S.D.).

Sensory Parameters	Ripening Time (Days)					
	60		90		180	
	TC	SST	TC	SST	TC	SST
Appearance/Colour (0–10)	9.14 ± 0.14 ^a^	9.33 ± 0.08 ^a^	9.21 ± 0.37 ^a^	9.17 ± 0.22 ^a^	9.42 ± 0.30 ^a^	9.28 ± 0.25 ^a^
Body/Texture (0–40)	35.14 ± 0.29 ^a^	35.14 ± 0.29 ^a^	36.76 ± 0.66 ^a^	36.67 ± 1.84 ^a^	34.89 ± 1.54 ^a^	33.56 ± 2.04 ^a^
Flavor/Odor (0–50)	42.98 ± 2.03 ^a^	44.52 ± 0.55 ^a^	45.48 ± 1.15 ^a^	45.12 ± 2.58 ^a^	40.00 ± 2.89 ^a^	39.86 ± 3.34 ^a^
Total (0–100)	87.26 ± 1.76 ^a^	89.00 ± 0.75 ^a^	91.45 ± 1.97 ^a^	90.95 ± 4.60 ^a^	84.31 ± 4.65 ^a^	82.69 ± 5.51 ^a^

^a^ Means in each parameter with the same superscripts for TC and SST values do not differ statistically significantly (*p* > 0.05) in each time interval.

## Data Availability

All related data and methods are presented in this paper. Additional inquiries should be addressed to the corresponding author.

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
