# Peer review of "Physicochemical, Microbiological and Sensory Characteristics of White Brined Cheese Ripened and Preserved in Large-Capacity Stainless Steel Tanks"

_foods, 2023, doi:10.3390/foods12122332_

Round 1

Reviewer 1 Report

The manuscript very well shows the ability to ripen the white brine cheese in stainless steel tanks. I suggest using the term ripening instead of ripening-preservation. The abstract should be improved. As well as the title. All comments and suggestions are given in the manuscript text attached. 

Minor English revision is needed.

Author Response

We thank you for your helpful comments and suggestions. We believe that have significantly strengthened our manuscript.

In the attached file below please find our response, point-by-point, to the comments.

Reviewer 2 Report

1.     I propose to modify the title of the manuscript: Physicochemical, microbiological and sensory characteristics of white brined sheep milk cheese ripening and preserving in large-capacity tanks

2.     Add some more information about conditions of sensory evaluation: average piece size, how many panelists were there in the study, sensory room temperature, lighting, distracting factors. Please add these informations to the point 2.4. In the future I propose to perform sensory analysis also after production.

3.     Why the number of lactic acid bacteria was not determined (look at the starter culture)?

Same more corrections are highlighted in the pdf file.

Author Response

We thank you for your helpful comments and suggestions. We believe that have significantly strengthened our manuscript.

In the attached file below please find our response, point-by-point, to the comments. please find our response, point-by-point, to the comments

Reviewer 3 Report

The objective of the present study was to investigate the effect of ripening and preservation containers on the characteristics of white cheese, ripened in brine. The comparison is made between large-capacity stainless steel tanks (SST) of 500 kg and the respective control samples on tin containers (TC) of 17 kg.

The research is interesting and the authors made a large number of analytical determinations to better highlight any differences between the two preservation/ripening systems. Therefore the paper is of scientific value. It is a pity that the statistical analysis carried out is not well explained in Materials and Methods (and perhaps it is not adequate, but it is not clear what the model is) and the text of the paper in the Results and Discussion is not precise and punctual, taking into account only of statistically significant differences. The authors continue to make speculative claims, accounting for differences not supported by statistics.

I therefore recommend rewriting Results and Discussion taking into account only the differences that are statistically significant. It would be even better to redo the statistical analysis taking into account not only the direct comparison between the two experimental theses, but also the “ripening time” as another fixed factor.

For all these reasons I suggest a Major revision.

Detailed comments:

Abstract: page 1 line 17: “…had higher salt concentration, hardness, elasticity, and gumminess.”. At the light of your statistical analysis, you cannot write this. Statistical differences between TC and SST are very few, and only those statistically significant.

Page 1 line 17: change “these ripened” with “those ripened”.

Page 1 line 33: “…used, , ripening”: delete one comma.

Page 1 line 40: delete a space between “as , Feta”.

Page 1 line 43: Add a space before “They”.

Page 2 line 51: Delete the point before [4-5]. Make this in all the paper (e.g. Page 2 line 69).

Page 2 line 53: “scalding process”: are you sure that the term “scalding” is correct?

Page 2 line 56: change “can be use” with “can be used”.

Page 2 line 62: change “however” with “but”.

Page 2 line 66: change “shape varies” with “shape it varies”.

Page 2 line 80: put the “2” of O2 in the subscript.

Page 2 line 81: you haven't yet defined the meaning of SST (the abstract doesn't apply). Define it here.

Page 2 line 62: too much space before “and the effect”.

Page 3 line 103: “bulgaricus” must be written without capital letter.

Page 5 line 143: put the “2” of N2 in the subscript.

Page 5 line 144: change “Fatty acids methylated” with “Fatty acids were methylated”.

Page 5 line 147: too much space before “Fatty acid methyl…”.

Page 5 lines 161-163: the sentence is written with different typographical characters.

Page 6 line 225: Add a comma between “Physicochemical” and “microbiological”.

Statistical analysis: you should better specify the statistical model that you have used as, for example (it is only an example, it is not your statistical model), Yijk = µ + Si + Cj + εijk  where: Yijk = dependent variable; µ = overall mean; Si = effect of §§§ (i = 1,…4; §§; §§; §§; §§); Cj =effect of §§ (j=1,….3; §§, §§, §§); εijk= residual error. The problem is that I do not understand if you have applied only one fixed factor (the treatment, i.e. TC or SST) or also the ripening time as fixed factor. I think that a correct statistical analysis should take into account of both the fixed factors. Moreover, the best one should be “repeated measures”. But I think that you have considered only the treatment, because in the tables, the letters of significance (a, b…) are only between the two treatment in the same ripening time for each ripening time. I am not an expert of statistics, but I think that in this way you completely lose some important information, i.e. the changing of the cheese during time. In the Result and Discussion section you have commented anyway, but if you have not statistical analysis you cannot write that the value at 10 d is different from that of 60 d, neither if the number is very different!!

Page 7 lines 244-247: you cannot make these comments: among different ripening times there is not statistics; anyway, between the two treatments there is the statistical analysis, but it tell you that there is no difference!! This is true also in a lot of other comments: line 255 (maximum), lines 258-259, lines 262-263, line 281 (higher), line 284 (slightly higher), line 287 (highest), line 308 (highest), lines 332-342 and so on. I think that all Results and Discussion should be rewritten taking into account this: only the statistical significant differences are differences; the others cannot ben commented.

Page 7 line 248: change “expressed as dry matter” with “expressed on dry matter”.

Page 7 line 273: change “Table 3 and 4 shows” with “Tables 3 and 4 show”

Page 8 line 299: change “PH” with “pH”.

Please, put the tables and the figure as close as possible to their description.

Moreover, there are too many lines. You should put only one horizontal line to separate the title “Ripening times (days)” from the line “TC – SST – TC –SST” and another line after this to separate this from the values. If you want, you can put another line, for example, in the Table 9, between the last fatty acid (C18:3 n3) and the line “CLA”. And the last line at the end of the table. All the other line should be deleted.

Moreover, Tables can be more large than the text. You should use this place to put all means and their SD in the same line, as for Table 6 and 7.

Page 12 line 416: change “Regarding their SFA” with “Regarding SFA”

Page 12 line 416: change “…and PUFA content were similar” with “…and PUFA content, they were similar”

Page 17 (in the manuscript, I don’t know why, it is renumbered again with 5. Correct it) lines 497-511. See above (Page 7 lines 244-247).

Page 17 (5) line 503: change “greater TVC than this of…” with “higher TVC than that of…”

Page 17 (5) line 517: the Table 14. Delete “the”

Page 17 (5) line 527: change “gradula” with “gradual”

Page 18 (6) lines 538-539: put a blank line between the note and the text.

Page 18 (6) lines 543-546: this sentence should be deleted: there are a lot of things that determine the quality of a cheese, not only fat and moisture. Moreover, the term “Of note” is not correct. It is better to write “It should be noted”.

Page 18 (6) line 552. There are two commas.

Literature is not well written according to the rules of “Foods”: in the paper from scientific journals, the name of the journal must be written in italics and also the number of the volume; the pages are normal written, and only the year is in bold.

Citation 27 is incorrect and must be completed.

The English language is not bad, but can be improved.

Author Response

(The authors gave the same response as above.)

Round 2

Reviewer 3 Report

I am sorry, but most of the modifications that I suggested were completely ignored by the authors. The authors continue to declare statings that are not supported by the statistical analysis. Also Tables were not corrected in the way I suggested. For this reasons this time I suggest to reject the paper. 

English language is improved from the last version of the manuscript, but there are still some imprecisions.

Author Response

Dear reviewer,

Thank you for your time reviewing our manuscript and providing valuable comments.  Your comments were useful and led to improvements in the current version. The authors have carefully considered the comments and tried our best to address every one of them. Please find attached our responses to your comments.

Round 3

Reviewer 3 Report

I noticed that the authors, this time, have made a great effort to follow and implement the suggestions that have been proposed. I think that the paper has now considerably improved, in particular as regards the discussion of the results: all references to comments on insignificant differences have been removed, and this is the most important modification, which makes the difference between a "reject" and a "minor revision". The statistical analysis was also clarified and well described, as I requested. Also literature is almost completely adjusted according to the Journal rules. In some places the English language could be improved, and I have suggested three sentences that should be modified, having incorrect English.  I therefore suggest a minor revision. 

Page 1 line 30: change “any significant statistically differences” with “any significant statistical difference”. 

Page 2 line 73: change “Their shape, it varies” with “The shape varies”. 

Page 9 line 349: change “compare” with “compared”. 

Page 9 line 350: change “cheese” with “cheeses”. 

Citation 10: change “Nega, A. ans Moatsou, G.” with “Nega, A.; Moatsou, G.”. 

Citation 16: change “Kaminarides, S. and Stachtiaris, S.” with “Kaminarides, S.; Stachtiaris, S.”. 

The use of “and” when there are only two authors is not suggested by the rules of the Journal. 

As for the tables, I should delete all horizontal lines that are not necessary (for example, I would leave one between individual fatty acids and fatty acid classes, such as SFA, PUFA, etc.), but Editors and Publishers will consider this aspect, which is only of an editorial nature, better than me.

/

Author Response

Dear Reviewer,

We again thank you for your additional helpful comments and suggestions.

Below please find our response, to your comments

Page 1 line 30: change “any significant statistically differences” with “any significant statistical difference”. 

The phrase “any significant statistically differences” has been changed with phrase “any significant statistical

Page 2 line 73: change “Their shape, it varies” with “The shape varies”. 

The phrase “Their shape, it varies” has changed with phrase “The shape varies”. 

Page 9 line 349: change “compare” with “compared”. 

The word “compare” has been corrected with the word “compared”

Page 9 line 350: change “cheese” with “cheeses”. 

The word “cheese” has been corrected with the word “cheeses”

Citation 10: change “Nega, A. and Moatsou, G.” with “Nega, A.; Moatsou, G.”. 

the citation 10 has been changed according  the rules of the Journal.

Citation 16: change “Kaminarides, S. and Stachtiaris, S.” with “Kaminarides, S.; Stachtiaris, S.”. 

the citation 16 has been changed according  the rules of the Journal.